# Extensions of realisations for low-dimensional Lie algebras

**Iryna Yehorchenko**⋆†

Institute of Mathematics, National Academy of Sciences of Ukraine, Ukraine
Institute of Mathematics, Polish Academy of Sciences, Poland

⋆ iyegorch@imath.kiev.ua , † iyehorchenko@impan.pl

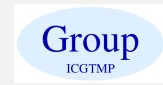

*34th International Colloquium on Group Theoretical Methods in Physics
Strasbourg, 18-22 July 2022*

## Abstract

**We find extensions of realisations of some low-dimensional Lie algebras, in particular, for the Poincaré algebra for one space dimension. Using inequivalent extensions, we performed comprehensive classification of relative differential invariants for these Lie algebras. We show difference between classification of extensions of realisations, and classification of nonlinear realisations of Lie algebras.**

## 1 Introduction

Work on classification of extended realisations of Lie algebras was motivated by the paper by M. Fels and P. Olver [1] that presents importance and need to have a procedure to classify Lie algebras that are used in various fields (see e.g. references therein). Mathematical relevance of such problem is its importance for the classical invariance theory and theory of special functions, and relevance in physics and practical applications, beyond quantum mechanics and computer vision, is the need to provide symmetry classification of mathematical models for different processes. Relative differential invariants, RDIs (not only absolute invariants, ADIs) shall be needed for full group classification of PDEs invariant under some Lie algebras. Classification of RDIs is an interesting problem of the abstract algebra by itself, even without any relevance to differential equations and their symmetry. RDIs were already applied for characterisation of some ordinary differential equations [2], and I believe that it is possible to find a method for characterisation of partial differential equations using differential invariants of equivalence groups. Extensions of realisations may be used not only for classification of RDIs, but in any problems of symmetry classification of models when we increase the set of variables but want to preserve symmetry of the initial model. Another application of RDIs is study of singularities of curves and manifolds.

The methods for full description and classification of absolute differential invariants are well-known. However, it appeared more difficult to find a way to classify RDIs. For more

information on the story behind this work see [3]. Some mathematicians (e.g. Oliver Glenn in [4]) claimed that they know how to classify the RDIs, but the reference [5] he cited does not contain the solution. The paper that inspired me to solve this problem [1] also did not contain a solution, and I presented my method of extended realisations of Lie algebras at the SPT-2007 conference [6]. Later I was informed by Pavel Winternitz that there exists an earlier alternative solution of this problem of classification of RDIs in their papers published in 1991 and 1993 [7] and [8] (these papers considered finding full sets of invariant differential equations, but their approach also completely solves a problem of classification of RDIs). However, I believe that the procedure proposed in [6] is much simpler. This procedure also allows to apply to RDIs all extensive theory for ADIs, e.g. ability to construct operators of invariant differentiation and fundamental bases of ADIs, enabling description of invariants of any order.

Here I present some new examples of classification of extended realisations that may be extended to higher-dimensional algebras. I also compare classification of extended realisations and classification of nonlinear realisations using an example of the Poincaré algebra for one space dimension, and show that these problems are different and produce different solutions.

Let us start with a realisation of a Lie algebra with the basis operators $L = \langle Q_m \rangle$,

$$Q_m = \zeta_{ms}(x_i)\partial_{x_s}, \tag{1}$$

where $x_i$ are some variables that may be regarded as dependent or independent in construction of some equations or differential invariants, and $\zeta_{ms}$ are some functions of $x_i$. In the examples there dependent variables are designated specifically, we may use other letters. Hereinafter we will imply summation over the repeated indices, if not specially indicated otherwise.

We take additional variables $R_k$, and study extended action operators $\hat{Q}_m = Q_m + \lambda_{mjk}R_j\partial_{R_k}$ that form the same Lie algebra with the same structural constants. These additional variables provide classification of symmetries for mathematical models in spaces extended with these variables, but preserving symmetries of original models. In specific application of classification of RDIs, these additional variables will be excluded from functional bases of ADIs of extended realisations to get bases of RDIs for the original realisations. For a specific realisation of any Lie algebra $L$, we can classify all inequivalent extended action realisations for a finite number of additional variables.

We consider construction of extensions for realisations of low-dimensional Lie algebras that were listed and classified in [9, 10]. Here we will deal only with one- and two-dimensional algebras, and the three-dimensional Poincaré algebra for one space variable with the aim to present main ideas. The problem of classification of the extended realisations (not only linear) is interesting by itself, but for classification of relative invariants of Lie algebras we need specifically only linear extensions with nonzero coefficients at $R\partial_R$.

**Definition 1**. A function $\Theta$ depending on $x, u$ and on partial derivatives $u$ of order up to $l$ ($\Theta$ may designate a set of functions $(\Theta_1, ...., \Theta_N)$) is called an RDI for the Lie algebra $L = \langle Q_m \rangle$, if it is an invariant of the $l$-th Lie prolongation of this algebra:

$$\overset{l}{Q}_m\Theta(x,u,\underset{1}{u},\ldots,\underset{l}{u}) = \lambda_m(x,u,\underset{1}{u},\ldots,\underset{l}{u})\Theta,$$

where $\lambda_m$ are some functions; if $\lambda_m = 0$, $\Theta$ is an absolute differential invariant (ADI) of the algebra; if $\lambda_m \neq 0$, it is a proper RDI.

**Definition 2**. A maximal set of functionally independent invariants of the order $r \leq l$ of a Lie algebra $L$ is a functional basis of differential invariants of the order $l$ for the algebra $L$.

Note that we cannot treat a set of independent RDIs the same way as we would treat a set of ADIs – a function of RDIs may be not invariant. E.g. linear combinations of RDIs generally speaking will not be RDIs. In the case when the task is to construct general extension operators

for operators (1), we look for the extensions in the form

$$\hat{Q}_m = Q_m + a_m(x_i, R)\partial_R.\qquad(2)$$

**Definition 3**. Extension of a realisation of the Lie algebra L with the basis operators of the form (1) is constructed of the same operators with more variables added (2).

To find scalar RDIs, we need to add just one new variable, and to find only linear extensions.

$$\hat{Q}_m = Q_m + a_m(x_i)R\partial_R.\qquad(3)$$

I used the lists of non-equivalent realisations of two-dimensional Lie algebras in [9, 10]. Please see the additional references there. We must note that the idea of equivalence/non-equivalence for the extended realisation accounts for classification of the RDIs - the difference with classification just with respect to the local transformation will be seen on the example of the translation operators. Our classification of the linear extensions has in mind the following definition of equivalence of RDIs.

**Definition 4**. Two RDIs of the algebra $L$ are called equivalent, if they can be transformed one into another by some transformation from the equivalence group of the relative invariance conditions.

$$\overset{l}{Q}_m R(x, u, \underset{1}{u}, \dots, \underset{l}{u}) = \lambda_m(x, u, \underset{1}{u}, \dots, \underset{l}{u})R.$$

We can also consider equivalence of pairs $(R, \lambda)$ of RDIs with their respective multiplicators. It may be useful for practical purposes of description of invariant differential equations, as a linear combination of RDIs with the same multiplicator will also be an RDI and may be used to construct an invariant differential equation.

The procedure for description of RDIs proposed in [6]: 1) Construct Lie prolongations of the operators $Q_m$; 2) write operators of extended action; 3) classify realisations of the extended action up to transformations from equivalence group of the invariance conditions; 4) find a functional basis of ADIs for the inequivalent realisations; 5) construct RDIs and ADIs of the algebra $L$ from absolute invariants of operators of extended action by elimination of ancillary variables. Note that ancillary variables $R$ may enter the ADIs of the operators of the extended realisation as multipliers of the form $R^K, K \neq 0$ are some integers - the ADIs having the form $FR^K$, where $F$ are some functions of dependent and independent variables, and derivatives of the dependent variables of the relevant order, will produce the RDIs of the form $F$.

Let us remind some properties of RDIs - a product of RDIs is also an RDI, an RDI in a non-zero degree will also be an RDI.

## 2 Extensions of algebras of translation operators

We start from a seemingly very easy case, that is a one-dimensional Lie algebra. It should be considered anyway for the purpose of comprehensive presentation. It is a Lie algebra whose basis consists of one operator. Any one separate first-order differential operator obviously forms a Lie algebra, and is locally equivalent to a translation operator.

Let us first construct a linear extension for the translation operator $\partial_x$. We will look for it in the form $\hat{Q} = \partial_x + R(x)F\partial_F$. It is easy to check that an arbitrary $R(x)$ will satisfy the commutator criterion for this algebra.

As to the standard classification up to equivalence with respect to the local transformations, we may use the Lie theorem on straightening out of vector fields as it was done e.g. in [11] (see [12]). However, though the operator $\hat{Q} = \partial_x + a(x)R\partial_R$ is certainly locally equivalent to $Q = \partial_x$, for our purpose of the RDI classification we need to consider operators with a

non-vanishing coefficient at $\partial_R$. It is easy to see that a non-extended $Q = \partial_x$ does not produce any RDIs, but the extended operator gave us an RDI $R = \exp x$. This RDI, as well as other exponential RDIs of the translation operators, is not useful at all to describe invariant equations, but it should be listed if we aim at obtaining a comprehensive classification of RDIs. So, the procedure of classification of RDIs requires classification of linear extended algebras with nonzero coefficients at $R\partial_R$.

Proper classification procedures are described e.g. in [13] and in references therein, as well as in the papers on classification of realisations of low-dimensional Lie algebras of the first-order differential operators [9, 10]. Such classification in the case of operators considered here shall find algebras or single operators whose actions are not equivalent under local transformations of the following form: $\tilde{x} = \kappa(x,R)$, $\tilde{R} = \phi(x,R)$, $\tilde{Q}_m = \tilde{a}_m(\tilde{x})\partial_{\tilde{x}} + \tilde{b}_m(\tilde{x})\tilde{R}\partial_{\tilde{R}}$. Similar criteria would be also relevant for algebras involving more variables. However, for the purposes of this paper we can easily check equivalence or non-equivalence using invariants of the relevant algebras or operators.

# 3 Two-dimensional Lie algebras

Using the procedure, proposed in [6], we classify extended realisations of the following two-dimensional Lie algebras

$$(a) \ \partial_x, \ x\partial_x, \qquad (b) \ \partial_x, \ y\partial_x, \qquad (c) \ \partial_x, \ x\partial_x + \partial_y, \tag{4}$$

excluding from our consideration here the algebras that consist only of the translation operators. The commutator of operators (4 (a)) is $\partial_x$. We consider the extension $\partial_x + a(x,y)R\partial_R$, $x\partial_x + b(x,y)R\partial_R$. From the commutation relations $b_x - xa_x = a$, we get determining conditions for the coefficients $-xa = \phi(y)$; $a$ is arbitrary. We get $b = xa(x,y) + \phi(y)$, and the general form of the extended operators $\partial_x + a(x,y)R\partial_R$, $x\partial_x + (xa(x,y) + \phi(y))R\partial_R$. Similarly we construct extensions for algebras (b) and (c).

Table 1

|   | Basis Operators | General Extended Basis Operators | Inequivalent Extended Basis Operators |
|---|---|---|---|
| 1 | $\partial_x, x\partial_x,$ | $\partial_x + a(x,y)R\partial_R,$ <br> $x\partial_x + (xa(x,y) + \phi(y))R\partial_R,$ | $\partial_x + R\partial_R,$ <br> $x(\partial_x + R\partial_R) + \epsilon R\partial_R,$ |
| 2 | $\partial_x, y\partial_x,$ | $\partial_x + a(x,y)R\partial_R,$ <br> $y\partial_x + (ya(x,y) + \phi(y))R\partial_R,$ | $\partial_x + R\partial_R,$ <br> $y\partial_x + (y + \epsilon)R\partial_R,$ |
| 3 | $\partial_x, x\partial_x + \partial_y,$ | $\partial_x + \Phi_x(x,y)R\partial_R,$ <br> $x\partial_x + (\Phi_y - x\Phi_x)R\partial_R,$ | $\partial_x + R\partial_R, \quad x\partial_x + R\partial_R.$ |

$a(x,y)$, $\phi(y)$, $\Phi(x,y)$ are arbitrary sufficiently smooth functions; $\epsilon$ is equal to 0 or 1.

Operators listed in Table 1 would allow calculation of zero-order RDIs. Finding higher-order RDIs would require finding extensions of the prolongations of operators of the initial realisation to the relevant order. We would like to point out that here we need extensions of the relevant prolongations of the operators being considered - not prolongations of extensions.

Here we will look for the **functional bases** of the RDIs up to the second order of derivatives. We find invariants for two independent variables $x$ and $y$, and one dependent variable $u$. Let us point out that finding differential invariants depends essentially from choice and assignment of dependent and independent variables.

Table 2

| | Extended Second Prolongations | Functional Bases of ADIs | Functional Bases of RDIs |
|---|---|---|---|
| 1 | $\partial_x + R\partial_R,\quad x\partial_x\ -u_x\partial_{u_x}$ $-u_{xx}\partial_{u_{xx}}\qquad -u_{xy}\partial_{u_{xy}}$ $+R\partial_R;$ | $y,\ u,\ u_y,\ u_{yy},\ u_x R,$ $u_{xx}R, u_{xy}R;$ | $y,\ u,\ u_y,\ u_{yy},\ u_x,\ u_{xx},$ $u_{xy},$ |
| 2 | $\partial_x\qquad\quad +\qquad\quad R\partial_R,$ $y\partial_x\ -\ u_x\partial_{u_y}\ -\ u_{xx}\partial_{u_{xy}}$ $-u_{xy}\partial_{u_{yy}} + R\partial_R,$ | $y,\ u,\ u_x,\ u_{xx},\ \exp\frac{u_y}{u_x}R,$ $u_x u_{xy} - u_y u_{xx},\ u_{xy}^2 -$ $2u_{xx}u_{yy},$ | $y,\ u,\ u_x,\ u_{xx},\ \exp u_y,$ $u_x u_{xy}\ -u_y u_{xx},\ u_{xy}^2 -$ $2u_{xx}u_{yy},$ |
| 3 | $\partial_x\ +R\partial_R,\qquad x\partial_x\ +\partial_y$ $-u_x\partial_{u_x}\qquad -u_{xx}\partial_{u_{xx}}$ $-u_{xy}\partial_{u_{xy}} +R\partial_R,$ | $\frac{\exp y}{R},\ u,\ u_y,\ u_{yy},\ u_x R,$ $u_{xx}R,\ u_{xy}R,$ | $\exp y,\ u,\ u_y,\ u_{yy},\ u_x,$ $u_{xx}, u_{xy}.$ |

# 4 Poincaré algebra for one space dimension

Many extensions were studied for many famous algebras of the mathematical physics without limitations of linearity and without any relation to finding RDIs (see e.g. [14] for the Poincaré algebra $P(1,2)$, and [11] for $P(1,1)$). These nonlinear realisations were used to find their differential invariants and whence new equations invariant under these algebras.

We will illustrate difference in the problem of classification of the general extensions for Lie algebras and of the extensions with the purpose of the RDI classification on the example of the Poincaré algebra with two independent variables $t$ and $x$ and one dependent variable $u$:

$$\partial_t,\qquad \partial_x,\qquad J = t\partial_x + x\partial_t. \tag{5}$$

A functional basis of the second-order ADIs can be written as follows [11]:

$$I_1 = u,\qquad I_2 = u_t^2 - u_x^2,\qquad I_3 = u_{tt} - u_{xx}, \tag{6}$$
$$I_4 = (u_t - u_x)^2(u_{tt} + 2u_{tx} + u_{xx}),\qquad I_5 = (u_t + u_x)^2(u_{tt} - 2u_{tx} + u_{xx}).$$

Setting $I_4$, $I_5$ to zero, the authors actually obtained expressions that are RDIs

$$AR_1 = u_t - u_x,\quad AR_2 = u_t + u_x,\quad AR_3 = u_{tt} + 2u_{tx} + u_{xx},\quad AR_4 = u_{tt} - 2u_{tx} + u_{xx}, \tag{7}$$

by means of listing invariant equations of the type $AR_i = 0$, but did not mention the concept of a relative differential invariant, and did not give any statements on full classification of such invariants.

Let us look at the extension of the standard realisation of the Poincaré algebra (5) constructed with the aim of classification of RDIs.

$$\partial_t + a(t,x)R\partial_R,\qquad \partial_x + b(t,x)R\partial_R,\qquad J = t\partial_x + x\partial_t + c(t,x)R\partial_R. \tag{8}$$

From the commutation relations $[P_t, P_x] = 0$, $[P_t, J] = P_x$, $[P_x, J] = P_t$ we get conditions on the functions $a(t,x)$, $b(t,x)$, $c(t,x)$: $a_x = b_t$, $c_t - ta_x - xa_t = b$, $c_x - tb_x - xb_t = a$, whence $a(t,x) = \Phi_t$, $b(t,x) = \Phi_x$, $c(t,x) = t\Phi_x + x\Phi_t + C$, where $\Phi = \Phi(t,x)$ is an arbitrary sufficiently smooth function of its arguments, and $C = const$.

Up to local equivalence and on condition of non-zero coefficients at $R\partial_R$, we obtain the following realisation

$$\partial_t + R\partial_R,\qquad \partial_x + R\partial_R,\qquad J = t(\partial_x + R\partial_R) + x(\partial_t + R\partial_R) + \epsilon R\partial_R, \tag{9}$$

where $\epsilon$ is equal to 0 or 1.

Operators (9) with $\epsilon = 0$ give from its functional basis of ADIs $R^{-1}\exp t$, $R^{-1}\exp x$ a set of RDIs $\exp t$, $\exp x$.

We can find an extended prolongation of realisation (5) using the commutation relations similarly, and obtain:

$$\partial_t + R\partial_R, \qquad \partial_x + R\partial_R,$$
$$J = t(\partial_x + R\partial_R) + x(\partial_t + R\partial_R) - u_t\partial_{u_x} - u_x\partial_{u_t} - u_{tt}\partial_{u_{xt}} - 2u_{xt}(\partial_{u_{xx}} + \partial_{u_{tt}}) - u_{xx}\partial_{u_{xt}} + \epsilon R\partial_R. \tag{10}$$

Operators (10) with $\epsilon = 1$ give from its functional basis of first-order ADIs

$$(u_t + u_x)R^{-1}, \qquad (u_t - u_x)R, \tag{11}$$

first-order RDIs for the algebra (5) $u_t + u_x$, $u_t - u_x$.

The determining equation for the second-order ADIs of the form $F = F(u_{tt}, u_{xt}, u_{xt})$ of (5), will look as follows: $2u_{xt}(F_{u_{xx}} + F_{u_{tt}}) + (u_{xx} + u_{tt})F_{u_{xt}} + RF_R = 0$. The resulting second-order ADIs are

$$u_{tt} - u_{xx}, \qquad (u_{tt} + 2u_{xt} + u_{xx})R^{-2}, \qquad (u_{tt} - 2u_{xt} + u_{xx})R^2. \tag{12}$$

In the same way we obtain two inequivalent proper second-order RDIs: $u_{tt} + 2u_{xt} + u_{xx}$, $u_{tt} - 2u_{xt} + u_{xx}$.

An invariant $u_{tt} - u_{xx}$ is an ADI of (5), so we do not include it into the list of proper RDIs.

Products of invariants in the relevant degrees from the lists (11), (12) to eliminate ancillary variables $R$ will give absolute invariants from the list (6). So to describe all non-equivalent RDIs up to the second order of the realisation being considered it is sufficient to take only zero-and first-order RDIs in addition to the list of ADIs.

The general extensions of the realisation (5) were studied in [11], and a new extended realisation was found:

$$\partial_t, \qquad \partial_x, \qquad J = t\partial_x + x\partial_t + u\partial_u, \tag{13}$$

that is not locally equivalent to (5), as well as ADIs of (13) up to the second order.

$$A_1 = u_t + u_x, \qquad A_2 = (u_t - u_x)u^{-2}, \qquad A_3 = (u_{tt} - u_{xx})u^{-1},$$
$$A_4 = (u_{tt} + 2u_{tx} + u_{xx})u, \qquad A_5 = (u_{tt} - 2u_{tx} + u_{xx})u^{-3}.$$

To obtain a comprehensive classification of RDIs for this extended realisation, we need to extend it further and to consider the realisation

$$P_t = \partial_t + a(t,x,u)R\partial_R, \qquad P_x = \partial_x + b(t,x,u)R\partial_R, \qquad J = t\partial_x + x\partial_t + u\partial_u + c(t,x,u)R\partial_R.$$

From the commutation relations $[P_t, P_x] = 0$, $[P_t, J] = P_x$, $[P_x, J] = P_t$ we get conditions on the functions $a(t,x,u)$, $b(t,x,u)$, $c(t,x,u)$:

$$a_x = b_t, \qquad c_t - ta_x - xa_t - ua_u = b, \qquad c_x - tb_x - xb_t - ub_u = a,$$

whence

$$a(t,x,u) = \Phi_t, \qquad b(t,x,u) = \Phi_x, \qquad c(t,x,u) = t\Phi_x + x\Phi_t + u\Phi_u + C,$$

where $\Phi = \Phi(t,x,u)$ is an arbitrary sufficiently smooth function, and $C = const$.

Up to local equivalence and on condition of non-zero coefficients at $R\partial_R$, we obtain

$$\partial_t + R\partial_R, \qquad \partial_x + R\partial_R, \qquad J = t(\partial_x + R\partial_R) + x(\partial_t + R\partial_R) + u\partial_u + \epsilon R\partial_R, \tag{14}$$

where $\epsilon$ is equal to 0 or 1.

Operators (14) with $\epsilon = 0$ give from its functional basis of ADIs $R^{-1}\exp t$, $R^{-1}\exp x$ a set of RDIs $\exp t$, $\exp x$.

We can find the extended prolongation of realisation (5) using the commutation relations similarly, and obtain:

$$
\begin{aligned}
&\partial_t + R\partial_R\,, \qquad \partial_x + R\partial_R\,, \\
&J = t(\partial_x + R\partial_R) + x(\partial_t + R\partial_R) + u\partial_u + u_x\partial_{u_x} + u_t\partial_{u_t} + u_{xx}\partial_{u_{xx}} + 2u_{xt}\partial_{u_{xt}} + u_{tt}\partial_{u_t} \\
&\qquad - u_t\partial_{u_x} - u_x\partial_{u_t} - u_{tt}\partial_{u_{xt}} - 2u_{xt}(\partial_{u_{xx}} + \partial_{u_{tt}}) - u_{xx}\partial_{u_{xt}} + \epsilon R\partial_R\,.
\end{aligned}
\tag{15}
$$

We can take relative differential invariants as follows:

$$
IR_1 = u\,, \qquad IR_2 = u_t + u_x\,, \qquad IR_3 = u_t - u_x\,, \qquad IR_4 = u_{tt} - u_{xx}\,,
$$
$$
IR_5 = u_{tt} + 2u_{tx} + u_{xx}\,, \qquad IR_6 = u_{tt} - 2u_{tx} + u_{xx}\,.
$$

## 5 Conclusion

We classified extended realisations for two-dimensional Lie algebras and for the Poincaré algebras for one space dimension, and found functional bases of absolute differential invariants for there new inequivalent realisations. These results allowed to classify RDIs for these realisations. It would be interesting to study RDIs for more algebras, and to look for new nonlinear realisations of such algebras.

## Acknowledgments

I would like to thank the Institute of Mathematics of the Polish Academy of Sciences for their hospitality and grant support, to the National Academy of Sciences of the USA and the National Centre of Science of Poland for their grant support.

I would like also to mention my good memories and appreciation of my visit to the Centre de recherches mathématiques in Montreal by invitation of late Jiří Patera where I had fruitful discussions with Jiří Patera and Pavel Winternitz, and I also was able to study literature on differential invariants, and, in particular, find papers by Oliver Glenn.

**Funding information**   The research and conference participation of the author were funded by Narodowe Centrum Nauki (Poland), Grant No.2017/26/A/ST1/00189.

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
