# Peer review of "Extensions of Realisations for Low-Dimensional Lie Algebras"

_SciPost Physics Proceedings, doi:SciPost Phys. Proc. 14, 048 (2023)_

## Round 1 · Referee Report · Anonymous (Referee 1) · 2023-4-14

Strengths

  1. Introduces a topic not so well known

  2. Includes some simple attractive examples

Weaknesses

  1. It is not clearly stated the originality of the results mentioned

  2. The work lacks of some motivation

  3. Applications should be given

Report

This paper deals on the extensions of realizations of Lie algebras and
their relative differential invariants (RDI).
It seems that this work is an introduction to this topic with some definitions
and a few examples including the one dimensional Poincar\'e Lie algebra of dimension three.

In general, this introduction to a topic which is not
so widely known is quite interesting. One reason, for not being so popular, could be that applications are not so clearly described or
enumerated. In this sense the author has taken the example
of the 1+1 Poincar\'e Lie algebra, which is so important in physics; but on the other hand we do not see what potential applications have the results here shown. We were expecting for instance, how to associate the extended realizations to a representation of the Lie group in a certain space.
Or to show the interest of the RDIs to describe some physical properties.
In order to introduce a theory related to physics, the search for applications is quite relevant to show its potential interest.

On the other hand, as an introduction the author should state clearly what are the new results included in this paper, and what are the most important results on the classification of extensions and their RDIs.

Requested changes

i) The few definitions should be more clear, or supplemented with some comments. For example Definition 4 about equivalent RDIs of a Lie algebra is given in terms of.... the equivalence of RDIs !!!

ii) It iss convenient to add some comments and/or explain some of the main applications of this topic, so that it would serve as a motivation for the reader.

iii) State clearly what is new and what are the known
results in this topic in this paper.

iv) There should be a more detailed explanation about why the extension term
$R\partial_R$ is so relevant. For instance by means of an example.

---

## Round 2 · Author Response

I have taken into account all the reviewer's comments and made the requested changes. I added more information on applications of relative differential invariants in mathematics and physics, and clarified which new and original results are presented in this paper. Definition 4 was clarified to avoid the impression of a circular definition. Examples and calculations in the paper are new results that was pointed out.

---

## Round 2 · List of Changes

1. New text of the Abstract.
2. Rewritten Introduction, including reformulation of Definition 4.
3. Added reference 3
4. Corrected mistake in line 4, column 4, Table 1 (page 4)
5. Rewritten Conclusions.
6. Changed the order of references in accordance with the rewritten introduction
7. Calculations in Section 3 on page 4 were abridged to leave space for extended motivation and application comments.

You are currently on this page

Resubmission scipost_202212_00054v2 on 19 June 2023

---

## Editorial Decision

published